# Hierarchically Porous Carbon Nanosheets from One-Step Carbonization of Zinc Gluconate for High-Performance Supercapacitors

**DOI:** 10.3390/ijms241814156

**Published:** 2023-09-15

**Authors:** Zhiwei Tian, Zhangzhao Weng, Junlei Xiao, Feng Wang, Chunmei Zhang, Shaohua Jiang

**Affiliations:** 1Jiangsu Co-Innovation Center of Efficient Processing and Utilization of Forest Resources, International Innovation Center for Forest Chemicals and Materials, College of Materials Science and Engineering, Nanjing Forestry University, Nanjing 210037, China; tianviatian@163.com (Z.T.); F2eyA123@126.com (J.X.); fengwang@um.edu.mo (F.W.); 2Strait Institute of Flexible Electronics (SIFE, Future Technologies), Fujian Normal University, Fuzhou 350117, China; 3Institute of Materials Science and Devices, School of Materials Science and Engineering, Suzhou University of Science and Technology, Suzhou 215009, China; cmzhang@usts.edu.cn

**Keywords:** zinc gluconate, ammonium chloride, porous carbon, supercapacitor, energy density

## Abstract

Supercapacitors, with high energy density, rapid charge–discharge capabilities, and long cycling ability, have gained favor among many researchers. However, the universality of high-performance carbon-based electrodes is often constrained by their complex fabrication methods. In this study, the common industrial materials, zinc gluconate and ammonium chloride, are uniformly mixed and subjected to a one-step carbonization strategy to prepare three-dimensional hierarchical porous carbon materials with high specific surface area and suitable nitrogen doping. The results show that a specific capacitance of 221 F g^−1^ is achieved at a current density of 1 A g^−1^. The assembled symmetrical supercapacitor achieves a high energy density of 17 Wh kg^−1^, and after 50,000 cycles at a current density of 50 A g^−1^, it retains 82% of its initial capacitance. Moreover, the operating voltage window of the symmetrical device can be easily expanded to 2.5 V when using Et_4_NBF_4_ as the electrolyte, resulting in a maximum energy density of up to 153 Wh kg^−1^, and retaining 85.03% of the initial specific capacitance after 10,000 cycles. This method, using common industrial materials as raw materials, provides ideas for the simple preparation of high-performance carbon materials and also provides a promising method for the large-scale production of highly porous carbons.

## 1. Introduction

Human beings are increasingly inclined to use clean electricity to replace oil resources, due to issues such as environmental pollution and the depletion of fossil resources [1,2]. Currently, higher and more stringent demands have been placed on energy storage devices, and battery-based energy storage systems have been fairly well developed [3,4,5,6,7]. However, battery-based energy storage systems are difficult to use in all application needs, due to their limited power density and poor cycle stability. Supercapacitors are a new type of energy storage device with rapid charge–discharge capabilities, featuring higher power density, a broader operating temperature range, and an exceptionally long lifespan [8,9]. Supercapacitors bridge the application gap between traditional capacitors and batteries, making them favored by many researchers; they have now become one of the hotspots for the development of new energy storage devices. At the current stage of research, the development of high-performance electrode materials and optimization of electrode fabrication strategies are considered to be one of the main directions for the future development of electrochemical energy storage technology.

Electrode materials for supercapacitors can generally be divided into metal oxides, conductive polymers, and porous carbon materials [10,11]. Carbon materials have attracted a lot of attention due to their ultra-long cycle stability and simple preparation methods [12]. Gluconate is a common type of industrial product that does not produce environmentally polluting gases or other impurities during the pyrolysis process. Through simple heat treatment, it can be transformed into porous carbon materials with a high specific surface area, which makes them an ideal carbon precursor. Fuertes et al. [13] prepared two-dimensional carbon nanosheets from sodium gluconate through a one-step heat treatment. These carbon nanosheets possess a high specific surface area, of 1390 m^2^ g^−1^. Notably, the material exhibits a specific capacitance of 140 F g^−1^ at a current density of 1 A g^−1^. Li et al. [14] utilized low-melting-point iron gluconate as a carbon precursor and combined it with KOH activation to synthesize hierarchical porous carbon nanosheets. These carbon nanosheets achieved impressive specific capacitances of 226 F g⁻¹ and 168 F g^−1^ at 1 A g^−1^ and 50 A g^−1^ current densities, respectively. Additionally, precursors such as magnesium gluconate [15] and cobalt gluconate [16] have been used for the preparation of supercapacitor electrodes. However, there are a large number of metal oxides in the materials obtained after high temperature pyrolysis of the above-mentioned gluconates (such as sodium gluconate, cobalt gluconate, iron gluconate, etc.). Therefore, post-processing requires tedious steps, such as pickling and washing, which greatly increase the preparation cost. In short, a simpler preparation method is urgently needed. It is worth noting that the pyrolysis of zinc gluconate is similar to that of zinc-containing metal–organic frameworks, where organic components are transformed into carbon during pyrolysis. The non-corrosive zinc metal element evaporates at high temperatures, leaving micropores on the carbon surface, which cannot be substituted by the aforementioned gluconates.

Although zinc gluconate powder can achieve a higher specific surface area during pyrolysis due to its self-activation effect, its poor electrochemical performance is attributed to its single-element composition. Nitrogen doping is a common strategy to improve the electrochemical performance of carbon materials by introducing heteroatoms into the carbon material framework [17,18]. In this study, an efficient and simple preparation method is adopted to synthesize N-doped porous carbon (ZnPCN-1) through one-step pyrolysis of uniformly mixed zinc gluconate and NH_4_Cl. ZnPCN-1 possesses a high specific surface area of 1162 m^2^ g^−1^ and a suitable nitrogen doping content of 4.57 at%. Consequently, ZnPCN-1 exhibits excellent electrochemical performance. The results show that a specific capacitance of 221 F g^−1^ is achieved at a current density of 1 A g^−1^. The assembled symmetrical supercapacitor achieves a high energy density of 17 Wh kg^−1^, and after 50,000 cycles at a current density of 50 A g^−1^, it retains 82% of its initial capacitance. Moreover, the operating voltage window of the symmetrical device can be easily expanded to 2.5 V when using Et_4_NBF_4_ as the electrolyte, resulting in a maximum energy density of up to 153 Wh kg^−1^, and it retains 85% of the initial specific capacitance after 5000 cycles.

## 2. Results and Discussion

The preparation process of ZnPCNs is illustrated in Figure 1a. Zinc gluconate and ammonium chloride are mixed uniformly through a simple dissolution method. The dried sample is then placed into a tube furnace for high-temperature pyrolysis. The organic part of zinc gluconate undergoes thermal decomposition, resulting in the generation of numerous micropores. Additionally, zinc elements evaporate directly at high temperatures, similar to zinc-based MOFs. As a result, the final product does not require acid washing. It should be noted that ammonium chloride generates a significant amount of NH_3_ and HCl gases at high temperatures, further activating the carbon material and introducing a certain amount of nitrogen heteroatoms into the carbon framework, which play a crucial role in enhancing the electrochemical performance of samples. SEM is a useful characterization to observe the surface morphology of materials [19,20,21]. The SEM morphology characterization of ZnPC, ZnPCN-0.5, ZnPCN-1, and ZnPCN-2 is shown in Figure 1b–e. It can be seen that all the samples show a sheet structure of different sizes, which is similar to other reports. In addition, the addition of ammonium chloride will cause carbon nanosheets to produce a large number of channels of nearly 100 nm, which will undoubtedly promote the transport of electrolyte ions. We speculate that ammonium chloride acts as a foaming agent during the pyrolysis process. With the gradual increase in NH_4_Cl proportion, the gas generated during the activation process increases, leading to a richer surface wrinkling and pore structure in the carbon, while still retaining the original two-dimensional layered structure. Figure 1f demonstrates that the carbon material exhibits a thin two-dimensional structure, consistent with the SEM results. It should be noted that a large number of microporous and mesoporous structures are distributed on the surface of the carbon material, suggesting that the carbon material should possess favorable electrochemical performance (Figure 1g). In general, the thin and porous two-dimensional nanosheet structure can effectively shorten the transmission distance of electrons and ions, and the large number of micropores generated by pyrolysis provides a large number of active sites for energy storage, providing a structural basis for the electrochemical performance of the material.

Figure 2a displays the XRD patterns of samples ZnPCN-0.5, ZnPCN-1, and ZnPCN-2. It is evident that all samples exhibit distinct broad peaks at around 23–25 and approximately 44 degrees, corresponding to the (002) and (100) crystal planes of the carbon, consistent with typical features of amorphous carbon [22,23]. Two peaks are observed in the Raman spectra of all samples, corresponding to the D-band and the G-band (Figure 2b) [24,25]. The I_D_/I_G_ ratios for ZnPCN-0.5, ZnPCN-1, and ZnPCN-2 are 1.58, 1.75, and 1.57, respectively. It should be noted that ZnPCN-1 has a higher value of I_D_/I_G_, possibly due to the formation of porous carbon material with enriched defect structures through NH_4_Cl activation and nitrogen doping in an appropriate proportion. This is expected to impart improved electrochemical performance to the material.

Figure 2c shows the nitrogen physical adsorption–desorption isotherms of the samples. It is evident that the curves of all samples exhibit type-I adsorption isotherms [26]. Typically, the isotherms rise sharply at low pressures due to the presence of numerous micropores in materials [27,28]. Additionally, a small hysteresis loop is observed at intermediate pressures, indicating a broad mesopore distribution in the structure. Overall, abundant micropores and mesopores exist in the porous carbon material after NH_4_Cl activation. The specific surface areas of ZnPCN-0.5, ZnPCN-1, and ZnPCN-2 correspond to 1135, 1162, and 1146 m^2^ g^−1^, respectively. ZnPCN-1 exhibits the largest specific surface area. Excessive NH_4_Cl may cause slight disruption to the pore structure of carbon, resulting in a decrease in specific surface area. As shown in Figure 2d, the samples exhibit similar pore size distributions, possibly because the addition of ammonium chloride does not significantly affect the micropore structure. Furthermore, the average pore diameters for ZnPCN-0.5, ZnPCN-1, and ZnPCN-2 are 3.32, 3.33, and 3.38 nm, respectively. Meanwhile, the pore volumes of the samples are 0.47, 0.49, and 0.48 cm^3^ g^−1^, respectively. Interestingly, the micropores of the carbon material provide adsorption sites for ions, and the mesopores are the electrolyte transport channels. ZnPCN-1 possesses a larger specific surface area and a suitable distribution of pore sizes. This pore structure provides favorable sites and channels for the storage and transport of electrolyte ions.

X-ray photoelectron spectroscopy (XPS) is a quantitative energy spectroscopy technology to determine the elemental composition, experimental formulas, and chemical and electronic states of the elements contained in materials [29,30,31]. Figure 2e presents XPS spectra of all the samples. It is evident that all samples contain C, O, and N elements, indicating successful nitrogen incorporation into the carbon framework. The nitrogen contents of ZnPCN-0.5, ZnPCN-1, and ZnPCN-2 are 2.95, 4.57, and 4.40 at%, respectively (Appendix A). ZnPCN-1 exhibits a higher nitrogen content, possibly due to excessive ammonium chloride hindering effective nitrogen doping. Figure 2f clearly shows the bonding configurations of different nitrogen-containing functional groups. The N 1s can be divided into pyridinic nitrogen (N-6), pyrrolic nitrogen (N-5), graphitic nitrogen (N-Q), and oxidized nitrogen by mathematical fitting, with corresponding binding energies of 398.3, 400, 401.5, and 405.6 eV, respectively (Figure 2f) [32]. N-6 and N-5 can enhance the wettability and specific capacitance of porous carbon, while graphitic nitrogen can improve the conductivity [33]. Furthermore, the C 1s can be divided into C=C/C-C, C-O/C-N, C=O, and O-C=O, with corresponding binding energies of 284.6, 285.3, 286.3, and 287.2 eV, respectively (Appendix A) [34]. The appearance of various of carbon-oxygen bonds and carbon-nitrogen bonds means that more defects are formed on the surface of carbon materials, which is crucial for high-performance electrochemical energy storage and good wettability. As shown in Appendix A, the O 1s spectrum can be divided into C=O, O-C-O, and O-H, with corresponding binding energies of 531.8, 532.7, and 533.7 eV, respectively. It is worth mentioning that the doping of oxygen atoms means that the material is more prone to defects and better electrolyte wettability [35]. In summary, ZnPCN-1 simultaneously possesses higher nitrogen and oxygen doping, providing a substantial pseudo-capacitance and significantly enhancing its electrochemical performance.

A three-electrode system was constructed using a 6 M KOH electrolyte to test the electrochemical performance of all samples. Figure 3a displays the cyclic voltammetry (CV) curves of all samples at a scan rate of 100 mV s^−1^. Clearly, ZnPCN-1 exhibits the largest enclosed area, indicating its superior electrochemical performance. Furthermore, ZnPCN-1 demonstrates the longest charge–discharge time (Figure 3b), suggesting its optimal specific capacitance, which aligns with the CV results. The CV curves of ZnPCN-1 at various scan rates are shown in Figure 3c. All CV curves exhibit rectangular shapes, suggesting favorable double-layer behavior. It is worth noting that, even at a high scan rate of 2000 mV s^−1^, the CV curve remains stable, indicating the remarkable rate capability of ZnPCN-1. Moreover, distinct galvanostatic charge–discharge (GCD) curves manifest symmetric isosceles triangles, reflecting favorable double-layer behavior and Coulombic efficiency (Figure 3d). The specific capacitances of ZnPCN-0.5, ZnPCN-1, ZnPCN-2, and ZnPC at a current density of 1 A g^−1^ are 178, 221, 188, and 80 F g^−1^, respectively. The variations of specific capacitance with current density for different samples are illustrated in Figure 3e. Undoubtedly, ZnPCN-1 exhibits the best specific capacitance and rate capability. The specific capacitance values of ZnPCN-1 correspond to 221, 198, 191, 188, 185, 179, and 174 F g^−1^ for current densities of 1, 2, 3, 4, 5, 10, and 20 A g^−1^, respectively. When comparing these with other previously reported carbon-derived electrodes (C9-250k-12 (197 F g^−1^) [36], MA6 (182 F g^−1^) [37], SAK (129 F g^−1^) [38], SSP-900 (199 F g^−1^) [39], Gna-CA (140 F g^−1^) [13]) (Appendix A), the electrodes used in this study possessed comparable or much better performances, suggesting the potential applications of ZnPCN-1 in practical fields in the future.

As shown in Figure 3f, the curves of all samples in the low-frequency region approach vertical lines, indicating good diffusion ability, favorable electron conductivity, and low internal resistance. A suitable equivalent circuit diagram is shown in Figure 3f, and the component values of the system are calculated according to the equivalent circuit. It should be pointed out that the R_CT_ and R_S_ values of ZnPCN-1 are 1.21 and 0.63 ohms, respectively, which are lower than those of other samples, indicating that it has the best electron conduction and ion transport properties. Worth mentioning is the remarkable cyclic stability of ZnPCN-1, retaining 96% of its initial capacitance after 50,000 cycles (Appendix A). Overall, the exceptional electrochemical performance of ZnPCN-1 is attributed to the suitable nitrogen doping and pore-forming effect of NH_4_Cl on zinc gluconate-derived carbon, introducing more active sites and significantly enhancing the electrochemical performance of the carbon material.

The reaction kinetics of the charge storage process can be evaluated by the CV curve. The total charge storage can be divided into capacitive effects and diffusion-controlled charge storage, which can be expressed by Appendix A [40]. In addition, the b value can be used as an important parameter to propose the kinetics of the electrochemical reaction of the material. In general, when the value of b is close to 0.5, it indicates that the energy storage process is mainly controlled by diffusion, and when the value of b is close to 1, this proves that the process is mainly controlled by capacitive effects. After calculation and fitting, the b value of ZnPCN-1 electrode is 0.82, which means that the energy storage process is mainly controlled by diffusion. Moreover, the capacitance contributions from the diffusion-controlled intercalation process and the surface capacitive effects can be quantitatively distinguished by Appendix A [40]. As shown in Appendix A, the diffusion-controlled reaction contributes 85% of the capacitance at a scan rate of 20 mV s^−1^, and it still contributes 47% of the capacitance at a scan rate of 500 mV s^−1^. This suggests that the thin and porous carbon nanosheets form a unique channel, which is conducive to the rapid transport of electrolyte ions, resulting in good electrochemical performance and rate performance of the obtained carbon materials.

The ZnPCN-1 was used as an electrode, with regular filter paper as the separator and 6 M KOH as the electrolyte, to assemble a symmetric supercapacitor device (ZnPCN-1//ZnPCN-1). As shown in Figure 4a, the CV curves exhibits a rectangular shape, indicating that the device displays favorable double-layer capacitance behavior. Moreover, even at a scan rate of 2000 mV s^−1^, the CV curve retains its rectangular shape, suggesting excellent rate capability. The charge–discharge curves at different current densities demonstrate symmetric isosceles triangle shapes (Figure 4b), indicating good double-layer charge–discharge behavior of the prepared carbon material electrode, consistent with the CV test results. Figure 4c demonstrates the specific capacitance of the device at current densities of 0.5, 1, 2, 5, 10, and 20 A g^−1^, corresponding to 124, 122, 116, 113, 111, and 100 F g-1, respectively. It is noteworthy that the device exhibits satisfactory rate performance, maintaining 85% of its initial capacitance, even at a current density of 20 A g^−1^. Additionally, the EIS plots confirm the low internal resistance and high ion transfer rate of this device (Appendix A). The device demonstrates excellent cycling stability at a high current density of 50 A g^−1^, retaining approximately 82% of its initial specific capacitance after 5000 cycles (Figure 4d). Furthermore, as shown in Figure 5d, the device achieves an energy density of 17.2 Wh kg^−1^ at a power density of 499 W kg^−1^. Even at a power density of 19 kW, the energy density remains as high as 13.4 Wh kg^−1^, demonstrating remarkable rate performance.

As is well-known, organic electrolytes can significantly broaden the potential window of supercapacitors, thereby enhancing the power density of these devices. The potential window of the ZnPCN-1//ZnPCN-1 device can easily be extended to 0–2.5 V when using Et_4_NBF_4_ electrolyte. Figure 5a displays CV curves of the device at different scan rates. All of the CV curves show a rectangular shape, indicating that it still exhibits good double-layer behavior in the Et_4_NBF_4_ electrolyte. Figure 5b shows the GCD curves, which all manifest symmetric isosceles triangle shapes, demonstrating favorable double-layer behavior. Interestingly, the Coulombic efficiency of the device is as high as 99.8%, even at a low current density of 0.5 A g^−1^, which means that it has a good prospects for applications. The specific capacitances of the device at current densities of 0.5, 1, 2, 5, 10, and 20 A g^−1^ correspond to 179, 178, 176, 175, 102, and 92 F g^−1^, respectively (Appendix A). The capacitance retention of the device is 51% at a current density of 20 A g^−1^. As shown in Figure 5c, even after 5000 cycles at a current density of 20 A g^−1^, the device still retains 85.03% of its initial specific capacitance, demonstrating good electrochemical cycling stability. As depicted in Figure 5d, the device achieves a high energy density of 153.4 Wh kg^−1^ at a power density of 1242 W kg^−1^. Furthermore, the device maintains an energy density of 70.93 Wh kg^−1^ even at an ultrahigh power density of 47 kW kg^−1^. As shown in Appendix A, the assembled devices exhibit excellent energy storage performance in both aqueous and organic electrolytes, and compare favorably with published supercapacitor work, with performance far exceeding devices such as WC-E-100-48//WC-E-100-48 (11.0 Wh Kg^−1^, 26.3 W Kg^−1^) [41], WBMs-800//WBMs-800 (9.4 Wh Kg^−1^, 227 W Kg^−1^) [42], Co(OH)_2_@CW//CW (6.5 Wh Kg^−1^, 236 W Kg^−1^) [43], NiCo-P//CW (12.1 Wh Kg^−1^, 395 W Kg^−1^) [44], N-M-O//Carbon (20.1 Wh Kg^−1^, 226 W Kg^−1^) [45], and MnOx/PANI//Carbon (30.7 Wh Kg^−1^, 800 W Kg^−1^) [46]. This means that the device has good application prospects.

In summary, the device assembled with ZnPCN-1 exhibits excellent electrochemical performance. Firstly, the introduction of nitrogen atoms gives the porous carbon more active sites, enhancing its electrochemical performance. Secondly, the activation effect of NH_4_Cl optimizes the pore structure of the porous carbon, enabling it to store more electrolyte and providing more rapid pathways for ion transport.

## 3. Materials and Methods

Materials: Zinc gluconate (98%) and NH_4_Cl (AR, 99.5%) were purchased from Sinopharm Group Chemical Reagent Co., Ltd., Shanghai, China. Acetylene black, polyvinylidene fluoride (AR), N, N-dimethyl pyrrolidone (AR, 99.5%), and Et_4_NBF_4_ (AR) were obtained from Hefei Cyber Co., Ltd., Nanjing, China.

Preparation of porous carbon materials: Zinc gluconate and NH_4_Cl were dissolved in deionized water, separately, with mass ratios of 2:1, 1:1, and 1:2. The resulting solutions were then transferred to a 70 °C oven and dried for 24 h. The obtained mixtures were placed in a tube furnace and heated at a rate of 5 °C/min under a nitrogen atmosphere to 950 °C, where they were held for 2 h. The resulting carbon materials were labeled as ZnPC, ZnPCN-0.5, ZnPCN-1, and ZnPCN-2, based on different mixing ratios.

Preparation of electrodes: Porous carbon, acetylene black and PVDF were fully stirred according to the mass ratio of 8:1:1, and then coated on nickel foam of about 1 cm × 1 cm, and the mass loading of each electrode was guaranteed to be about 2 mg. Subsequently, the nickel foam was placed in a vacuum drying oven at 80 °C for 12 h. Finally, the dried nickel foam was placed in a tablet press to obtain the electrode at a pressure of 10 MPa.

Characterizations: The microstructure of the samples was characterized by X-ray diffraction (XRD, Rigaku D/Max 2500, Tokyo, Japan), field-emission scanning electron microscopy (SEM, Hitachi, FESEM, S-3400, Tokyo, Japan), and transmission electron microscopy (TEM, JEM-2010EX, Tokyo, Japan). The pore structure of obtained samples was examined via N_2_ adsorption/desorption experiments at 77 K using a micromeritics apparatus (BeiShiDe Instrument-S&T 3H-2000PS2, Beijing, China). The specific surface area was calculated by the Brunauer–Emmett–Teller (BET) method, and the pore size distribution and pore volume were calculated from the BJH model. All the samples were degassed under vacuum at 200 °C for 6 h before testing. Raman spectra were collected from a Raman spectrometer (Jobin Yvon, HR800, Paris, France).

Electrochemical measurements: Cyclic voltammetry (CV), galvanostatic charge–discharge (GCD), electrochemical impedance spectroscopy (EIS), and amperometry measurements were performed using a CHI760D electrochemical workstation (Chenhua, Shanghai, China). In addition, all EIS data were verified by Kramers–Kronig residual analysis to ensure the reliability of the obtained data. Briefly, we set the samples as the work electrode, the Pt foil and the Hg/HgO electrode as the counter and reference electrodes, respectively, and immersed them in a 6 M KOH electrolyte solution to form a three-electrode cell operated at room temperature (Appendix A). In addition, two electrodes of similar quality were selected as the positive and negative electrodes, and the filter paper was used as the separator. The 6 M KOH and 1 M Et_4_NBF_4_ were used as the electrolyte to assemble and package the CR2032 button cell. The specific capacitance of three-electrode system (*C_m_*, F g^−1^), the specific capacitance of double-electrode system (*C_s_*, F g^−1^), energy density (Wh Kg^−1^), and power density (W Kg^−1^) were estimated from GCD using the following Equations (1)–(4):(1)$$C_{m}=\frac{I\times \Delta t}{m\times \Delta V}$$
(2)$$C_{s}=2\times \frac{I\times \Delta t}{m\times \Delta V}$$
(3)$$E=\frac{1}{2}C\Delta V^{2}$$
(4)P=EΔt
where *I* (mA) is the rate of current, Δ*t* (s) is the discharge time, Δ*V* (V) is the voltage drop, and *m* (mg) is the mass of active material in working electrode.

## 4. Conclusions

This study uses an efficient and simple preparation method, wherein zinc gluconate and NH_4_Cl are uniformly mixed, followed by one-step pyrolytic synthesis to create nitrogen-doped porous carbon (ZnPCN-1). ZnPCN-1 exhibits a high specific surface area of 1162 m^2^ g^−1^ and a suitable nitrogen doping content of 4.57 at%. The addition of ammonium chloride not only optimizes the pore structure of the carbon but also introduces nitrogen heteroatoms that enhance the electrochemical performance of carbon. As a result, ZnPCN-1 demonstrates outstanding electrochemical properties. The results indicate a specific capacitance of 221 F g^−1^ at a current density of 1 A g^−1^. The assembled symmetric supercapacitor achieves a high energy density of 17 Wh kg^−1^, and even after 50,000 cycles at a current density of 50 A g^−1^, it retains 82% of its initial capacitance. Furthermore, when using Et_4_NBF_4_ as the electrolyte, the operational voltage window of the symmetric device can easily be extended to 2.5 V, resulting in an energy density of up to 153 Wh kg^−1^, while maintaining 85% of the initial specific capacitance after 5000 cycles. This method employs inexpensive industrial materials and involves only dissolution and a one-step pyrolysis process, providing a new concept for the simple and green preparation of high-performance carbon materials.

## Figures and Tables

**Figure 1 ijms-24-14156-f001:**
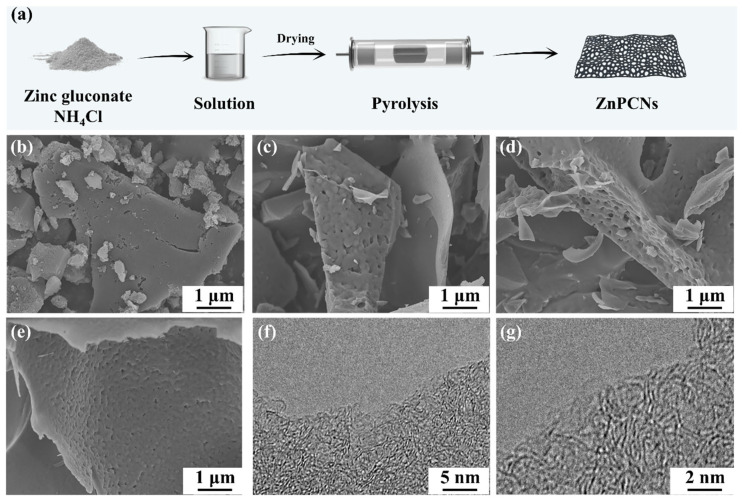
Schematic of the preparation process of the ZnPCNs (**a**). SEM diagrams of the prepared sample: ZnPC (**b**), ZnPCN-0.5 (**c**), ZnPCN-1 (**d**), ZnPCN-2 (**e**). TEM diagram of ZnPCN-1 (**f**,**g**).

**Figure 2 ijms-24-14156-f002:**
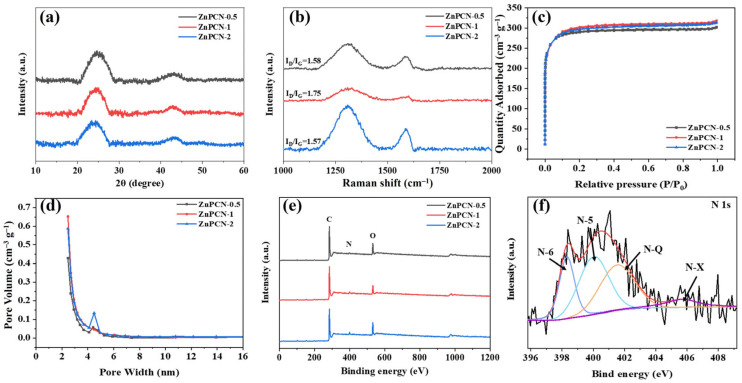
Structural characterizations of samples: XRD patterns (**a**), Raman spectra (**b**), Nitrogen adsorption–desorption isotherms (**c**), pore size distribution curves (**d**), XPS survey spectra (**e**), and high-resolution of N 1s (**f**) of ZnPCN-1.

**Figure 3 ijms-24-14156-f003:**
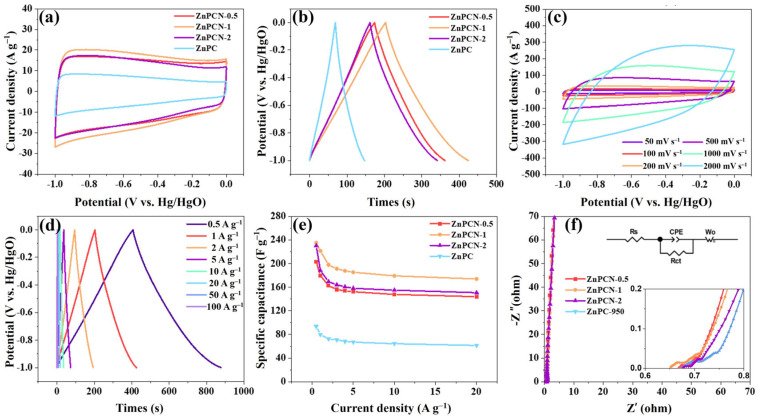
Electrochemical performance tested in a three-electrode system with 6 M KOH as the electrolyte: CV curves (**a**) of the electrodes at a scanning rate of 100 mV s^−1^ and GCD curves (**b**) measured at a current density of 1 A g^−1^ of samples; CV curves (**c**) at different scanning rates and GCD curves (**d**) at different current densities; rate capability curves (**e**) and EIS plots (**f**) of samples.

**Figure 4 ijms-24-14156-f004:**
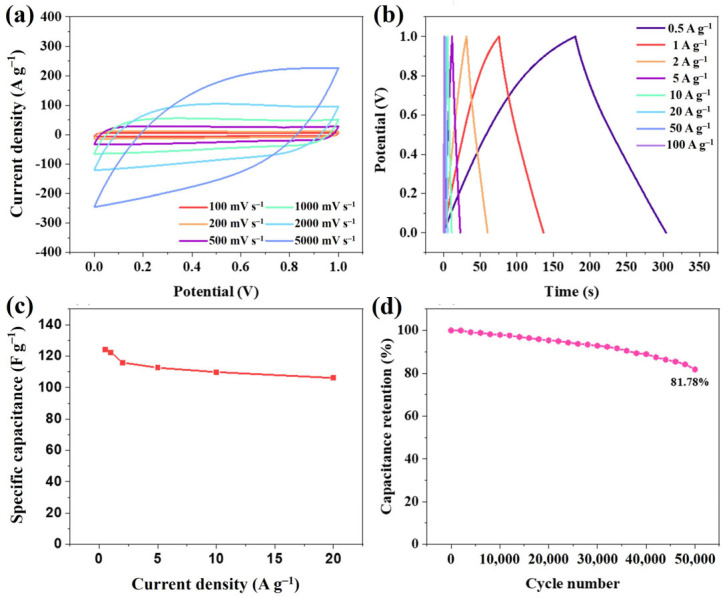
Electrochemical performance of ZnPCN-1//ZnPCN-1 in 6 M KOH: CV curves at different scanning rates (**a**), and GCD curves at various current densities (**b**); rate performance (**c**), and cycle performance curve for 50,000 cycles at the current density of 50 A g^−1^ (**d**) of the SSC device.

**Figure 5 ijms-24-14156-f005:**
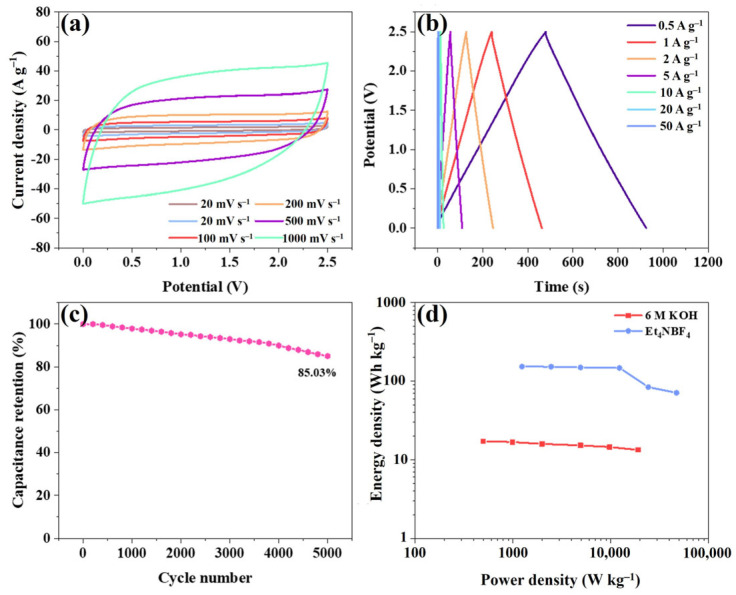
Electrochemical performance of ZnPCN-1//ZnPCN-1 in Et_4_NBF_4_ electrolyte. CV curves at various scanning rates (**a**), GCD curves at different current densities (**b**), cyclic performance for 50,000 cycles at the current density of 20 A g^−1^ (**c**), and Ragone diagram (**d**) of the SSC device.

## Data Availability

The data presented in this study are available on request from the corresponding author. The data are not publicly available due to the requirements of our further research.

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
