# Peer review of "Hierarchically Porous Carbon Nanosheets from One-Step Carbonization of Zinc Gluconate for High-Performance Supercapacitors"

_ijms, 2023, doi:10.3390/ijms241814156_

Round 1

Reviewer 1 Report

Dear Authors,

Congratulation on putting up a good manuscript draft. I have a few minor comments:

1. It would be good if you manage to put the graphs/images/cartoons close to the discussion like line 105 discusses XRD characterisation but the patterns appear at the end of the next page.

2. I think the introduction section is lacking in depth and perspective. I suggest improving it by comparing your work to recently published work. You can do this by discussing the alternative importance of zinc gluconate and ammonium chloride materials and comparing it with other commonly used materials. The following work by Dr Dar is suggested for reading and citing, https://doi.org/10.1016/j.jallcom.2023.170523. Also have a look at this work, https://doi.org/10.1038/nmat2297

3. Also please provide the details of the Kramers–Kronig residual analysis for the EIS data validation or add them as SI to the manuscript.

Best

Reviewer 2 Report

Reviewer’s comments

Manuscript Number: ijms-2590259

Title: Hierarchically Porous Carbon Nanosheets from One-step Carbonization of Zinc Gluconate for High-Performance Supercapacitors

1.          EIS data (Fig. 3f) should be fitted using an equivalent circuit and the fitting results should be displayed together with the experimental data.

2.          The correlation between structural/morphological findings and electrochemical performance should be discussed and compared with other related composites.

3.          Elemental composition analysis (EDX, ICP, etc.) is required to identify the material composition.

4.          Trasatti’s analysis is recommended to be done to analyse the diffusion and capacitance contribution.

Minor editing of English language required

Reviewer 3 Report

1-The name of the reference electrode is missing in the CV and GCD.

2- figures for the three-electrode graph.

3-What is the loading mass of the carbon electrode?

4-Include equations for specific capacitance, energy density, and power density.

5-Specify the purity of the chemicals used.

6-Incorporate the Coulombic efficiency.

7-Include the equivalent circuit in the EIS analysis.

Minor editing of English language required

Reviewer 4 Report

The manuscript titled “Hierarchically Porous Carbon Nanosheets from One-step Carbonization of Zinc Gluconate for High-Performance Supercapacitors” by Tian et al. has been reviewed where the authors have reported the preparation of three-dimensional hierarchical porous carbon materials with high specific surface area and suitable nitrogen doping using a one-step carbonization strategy. Here the materials they have used are the common industrial materials and has the large-scale production feasibility of porous carbon materials. Therefore, this work has significance in the light of its possible use in industry and hence can be accepted for publication once the following concerns are addressed.

1.       Line No. 50: B. Fuertes et al. to be written as Fuertes et al.

2.       Since the authors have claimed the developed material as hierarchically porous nanosheets, hence further characterization to demonstrate porosity, size-distribution of the pore diameters, sheet-like morphology, nano-dimension of the sheets, etc. would be necessary to support their claim.

3.       Some SEM images with low scale bar would be necessary to have a better look at the morphology and porous nature of the material.

4.       Including a table to compare the data obtained in this study with the reported studies will help understand the superiority of the material designed in this study.

Thus, I would like to invite the authors to address these points to further improve the scientific merit of this work. At this point the characterization part seems to be improved and at the same time the results should be compared with the available data published in the literature to better understand the novelty of the work.

The quality of the English laguage is acceptable.

Reviewer 5 Report

In their work, the authors have synthesized 3D hierarchical porous carbon materials with high specific surface area and suitable nitrogen doping using zinc gluconate and ammonium chloride as precursors. The Porous carbon used as an a supercapacitor with outstanding electrochemical properties. The results and the methods are well-presented, the discussion needs to be more developped.

- What was the effect of Oxygen and nitrogen content on the electrochemical properties, correlations of the effect of each heteroatom with the capacitance behaviour should be added

-  Figure S1 (a), correct Y axis

- It is hard to see the charge transfer resistance from figure 3.f. zoom on the high frequency region

- All the potentials in the figures should be references ( V vs Hg/HgO)

Round 2

Reviewer 2 Report

The authors tried to address most of the comments in the revised version. The current version could be accepted for publication. 

 Minor editing of English language required

Reviewer 4 Report

Although the authors have not provided the SEM images at a low scale, which they would have done at least for the reviewer to see and confirm, the manuscript seems improved and may be considered for acceptance. 

Minor grammatical corrections are required.